# The Impact of the COVID-19 Pandemic on Chinese Postgraduate Students’ Mental Health

**DOI:** 10.3390/ijerph182111542

**Published:** 2021-11-03

**Authors:** Zhengyan Liang, Derong Kang, Minqiang Zhang, Yuanlin Xia, Qing Zeng

**Affiliations:** 1School of Psychology, South China Normal University, Guangzhou 510631, China; 2020010220@m.scnu.edu.cn (Z.L.); 2020023477@m.scnu.edu.cn (D.K.); xiam13@126.com (Y.X.); 2020023457@m.scnu.edu.cn (Q.Z.); 2Key Laboratory of Brain, Cognition and Education Sciences, Ministry of Education, South China Normal University, Guangzhou 510631, China; 3Center for Studies of Psychological Application, School of Psychology, South China Normal University, Guangzhou 510631, China; 4Guangdong Key Laboratory of Mental Health and Cognitive Science, South China Normal University, Guangzhou 510631, China

**Keywords:** COVID-19, postgraduate student, mental health, anxiety, depression, social anxiety

## Abstract

To understand the mental health status of Chinese postgraduate students during the COVID-19 pandemic, we used three online questionnaires: self-rating anxiety (SAS) scale, self-rating depression (SDS) scale, and social avoidance and distress (SAD) scale. A total of 3137 postgraduate students from different regions of China participated in our study. We explored the relationship between participant characteristics and mental health using an analysis of variance (ANOVA). We found that the proportions of respondents with severe, mild, and moderate depression were 1.4%, 10.48%, and 21.99%, respectively, and the corresponding proportions of respondents with anxiety were 1.56%, 4.65%, and 14.69%, respectively. A one-way ANOVA revealed that the mental health statuses of the participants were different between the subgroups based on majors, classes, degree types, and the method of communication with advisors and students. A two-way ANOVA revealed significant effects on interaction and the method of communication with advisors and peers. These findings suggest that the mental health of postgraduate students should be monitored during the pandemic, especially when they are unable to communicate directly with their advisors or peers, and targeted psychological counselling must be focused on anxiety and depression.

## 1. Introduction

Since the outbreak of COVID-19 at the end of 2019, the normalcy of people’s lives has not been fully restored. Because this disease is highly contagious, isolation is one of the most effective measures to prevent the spread of COVID-19, and long periods of isolation, semi-segregation, or social control have a huge impact on various aspects of people’s lives, such as learning and work. 

To stop the spread of the pandemic, educational institutions need to implement closure measures or move some courses online. As of 1 April 2020, the number of learners required to stay at home due to the closure of their educational institution on all levels reached a peak of 1.598 billion across 194 countries [1]. In response to the changes brought by the closure of educational institutions due to the pandemic, a large number of scholars conducted research on college students in their respective countries.

Studies show that the impact of the pandemic and the lockdown measures on students in higher education is manifested in three ways.

First, there are changes in learning patterns, e.g., online courses, library closures, altered communication channels for teacher and administrative support, new assessment methods, etc. [2,3,4,5]

Second, there are significant changes in social patterns, such as home study; no normal contact with friends, classmates or relatives; no social events; no travel; stranded abroad and difficult to return home, etc. [2,6,7,8,9]. 

Third, there is the crisis in lifestyle and mental health. For lifestyle, including the reduction in physical activity, more sedentariness [10,11]; a decrease in sleep time [12,13]; changes in dietary behaviors [7]; negative emotions caused by the threat of COVID-19; as well as discomfort caused by lockdown measures, such as fear, frustration, stress, social isolation, anxiety, depression, etc. [2,6,14,15].

The student population in China is large, and the total number of students at all levels is 289 million, which accounts for 20.14% of the total population. Among them, the total number of students enrolled in higher education is 41.83 million, including 466,500 doctoral students and 2.673 million master students [16]. Meanwhile, lockdown measures in China are considered to have had positive results at public health, economic and administrative levels [17,18], and its role at the psychological level is also studied by scholars. Zheng et al. found that lockdown measures buffered the detrimental effect of the COVID-19 pandemic severity on anxiety [19]; Chang et al. found that Chinese college students had higher detection rates of anxiety and depression during lockdown [20]. Except China, scholars in Malaysia [21], Bangladesh [22], Greece [23], Spain [24], and India [25,26] found that respondents reported more severe psychological problems such as anxiety, depression, stress, fatigue, etc., during the lockdown. In view of the above, most studies show the negative impact of lockdown measures on the mental health of college students. It is necessary to pay more attention to this issue and to explore the relationship between psychological problems and demographic characteristics, so as to provide assistance for targeted psychological intervention. 

Among the higher education groups, postgraduate students studying in Chinese universities were selected as research subjects, mainly based on the following reasons: 

(1) The Chinese postgraduate group totaled 3,135,500 [16], while the domestic research for this group had a small sample size and a limited regional and national coverage [20,25,26,27,28]. (2) Postgraduate students have certain peculiarities compared to undergraduate students. From the demographic characteristics, postgraduate students are older, have studied for longer, and have a steady source of income; from a psychological point of view, postgraduate students have more serious psychological disorders than undergraduate students [29] and face greater learning challenges and schoolwork pressures [30]. Yet, postgraduate students are also better able to align their experiences and plans, adjust their goals, and demonstrate a greater mental resilience [31]. Therefore, this study conducted a survey on postgraduate students in Chinese universities to understand their mental activity and mental health status in the context of the COVID-19 pandemic and the normalization of its prevention and control, in order to provide an important reference for targeted psychological management. An analysis of the impact of change can provide a basis for improving future online learning and distance learning.

## 2. Materials and Methods

### 2.1. Participants

This survey was organized and implemented by Professor Minqiang Zhang and his team from the School of Psychology, South China Normal University. A total of 3137 postgraduate students (93.4% valid response rate) were selected in snowball sampling from 33 provinces or autonomous regions in China via the popular Chinese professional survey website Wenjuanxing (www.sojump.com; accessed on 7 February 2020). All participants were recruited by a widely used social platform (i.e., WeChat) by online fliers, which directed them to the online survey website (www.sojump.com; accessed on 7 February 2020). Each of them was first informed about the study online and they submitted an informed consent before starting the study. All participants completed the survey within 10 min on 7 February 2020. The sample included 672 male students (21.42%) and 2465 female students (78.58%); 2829 postgraduate students (90.18%) and 308 doctoral students (9.82%); and 1442 academic postgraduates (45.97%) and 1695 professional postgraduates (54.03%).

### 2.2. Data Analysis

In this study, we used Rstudio version 1.0.136 (Rstudio Corporation, Boston, MA, USA) for data analysis. Categorical variables are descriptive statistics, and continuous variables are expressed as mean ± standard deviation (M ± SD); we used *t*-test and one-way analysis of variance (ANOVA) to determine the differences in mental health statuses between participant characteristic-based groups, and two-way ANOVA was used to explore the main factors influencing depression, anxiety, and social anxiety and to classify anxiety and depression levels based on Chinese norms.

### 2.3. Measures

#### 2.3.1. Self-Rating Anxiety Scale (SAS)

The Self-Rating Anxiety Scale was used to assess people’s anxiety during the last week [32]. The scale included 20 self-evaluation items, from which 5 items were scored in the reverse order. The scale was a 4-item Likert scale with scores on responses ranging from 1 (occasionally) to 4 (always). The sum of the item scores was referred to as the total rough score, and the total rough score was multiplied by 1.25 to obtain the standard score. The Chinese version of the scale had a Cronbach’s alpha equal to 0.85. The norm of SAS in Chinese is: 50−59 scores indicated mild anxiety; 60−69 scores indicated moderate anxiety; and scores > 69 indicated severe anxiety.

#### 2.3.2. Self-Rating Depression Scale (SDS)

The Self-Rating Depression Scale was used to assess people’s depression during the last week [33]. The scale included 20 self-evaluation items, from which 5 items were scored in the reverse order. The participants responded on the 4-item Likert scale with scores ranging from 1 (occasionally) to 4 (always). The total rough score was the sum of scores of all items, and the total rough score was multiplied by 1.25 to obtain the standard score. The Chinese version of the scale had a Cronbach’s alpha equal to 0.88. The norm of SAS in Chinese is: (53−61) scores indicated mild depression; (62−71) scores indicated moderate depression; and scores > 72 indicated severe depression. 

#### 2.3.3. Social Avoidance and Distress Scale (SAD)

The Social Avoidance and Distress Scale was used to assess people’s anxiety during the last week [34]. The scale included 28 true–false items. Social avoidance and distress, respectively, referred to the tendency to avoid social interaction and the feeling of distress when experiencing such situations. Evasion is a manifestation of behavior; distress is an emotional reaction. The scale contained 28 self-assessment items, 14 of which were avoidance subscales and 14 of which were anxiety subscales. Each item was scored in a “yes-no” manner, with “1” indicating yes and “0” indicating no. The Cronbach’s α coefficient for SAD in this study was 0.921. 

The three scales used in this study essentially maintain their original versions, and only the sentences of individual items are revised according to the current context to ensure that the subjects understand the meaning of each item. The test for the common method deviation using the Harman one-way test means that the first factor has an interpretation rate of variance of 39.4%, not 50%, so there is no common method bias.

## 3. Results

### 3.1. Anxiety and Depression

A total of 3137 postgraduate students participated in this study. The descriptive results of the self-rating anxiety (SAS) scales and self-rating depression (SDS) scales are shown in the table below. Table 1 shows that the postgraduate anxiety scores and depression scores during the pandemic were higher than the Chinese norms, and the differences were highly statistically significant (*p* < 0.001).

According to the Chinese criteria, 656 (20.91%) and 1063 (33.88%) postgraduate students were identified as having symptoms of anxiety and depression, respectively. Specifically, the incidence rates of mild, moderate, and severe anxiety were 14.69%, 4.65%, and 1.56%, respectively, and those of mild, moderate, and severe depression were 21.99%, 10.48%, and 1.4%, respectively.

Prior to the outbreak, some researchers had studied the utility of SAS and SDS for postgraduate students: Wu [35] used SAS and SDS to evaluate anxiety and depression in 177 postgraduate students and found that the corresponding rates were 10.7% and 24.3%, respectively. Wang [36] conducted a survey on 941 students pursuing a professional postgraduate degree and found that the detection rates for both anxiety and depression symptoms were 2.6% and 8.5%, respectively, both of which were lower than the rates observed in the current study. With regard to other studies on the mental health status of Chinese college students during the COVID-19 pandemic, Chang [20] conducted a survey on 3881 students in Guangdong University and found that the incidence of anxiety among university students was 26.60%; Mao et al. [37] found depression and anxiety symptoms in 38.75% and 24.17%, respectively. Other researchers detected a higher rate of anxiety and depression in postgraduate students during the pandemic than before the pandemic (see Table 2).

### 3.2. Social Anxiety 

The survey used the social avoidance and distress (SAD) scale to measure the social anxiety conditions of 3137 postgraduate students. As shown in Table 3, the total and subscale (two subscales) scores were higher than the Chinese norms (*p* < 0.001). This suggests that the disease control measures employed during the pandemic have a serious impact on the social experience of postgraduate students and caused a high level of social anxiety.

### 3.3. Comparison of Mental Health Statuses of Postgraduate Students

The group differences between scale scores and the associations between participant characteristics and the scale scores examined using a *t*-test and ANOVA are presented below (Table 4). 

#### 3.3.1. Gender-Based Differences

The *t*-test showed that there were no significant, gender-based differences in the anxiety, depression, and social anxiety scores of postgraduate students (see Table 4) Researchers such as Jiang [38] conducted a meta-analysis on the mental health of postgraduate students and concluded that the impact of gender on the mental health of postgraduate students was not significant. Our findings are consistent with those of most previous studies, and these findings indicate that regardless of gender, the mental health of every postgraduate student is seriously affected by the pandemic, and the psychological problems of all students should be given equal attention.

#### 3.3.2. Degree Type-Based Differences

The *t*-test showed that there was a significant degree of type-based differences in anxiety scores (t = 6.15, *p* = 0.013 [<0.05]) and social anxiety scores (t = 8.04, *p* = 0.004 [<0.05]), but not in depression scores (t = 3.39, *p* = 0.066 [>0.05]) (see Table 4). The anxiety scores and social anxiety scores were significantly higher for academic postgraduate students than for professional postgraduate students. The differences in academic disciplines and in the requirements of academic support may have led to the differences between these groups.

#### 3.3.3. Professional Background-Based Differences

The ANOVA analysis (see Table 4) showed that there were significant, professional, background-based differences in the depression scores (F = 5.65, *p* = 0.004 [<0.05]); the highest postgraduate depression score was observed among students with a liberal arts background (48.98 ± 11.53), which is far higher than the norms of Chinese depression score (41.88 ± 10.57). Students with liberal arts backgrounds are more sensitive and have a strong ability to sympathize; this may be the reason behind the high severity of their depressive symptoms. Nonetheless, no significant, professional, background-based differences in the anxiety scores and social anxiety scores were found among postgraduate students, which meant that the professional background did not affect anxiety and social anxiety.

#### 3.3.4. Teaching Method-Based Differences

The ANOVA analysis (see Table 4) showed that there were significant teaching method-based differences in the anxiety scores (F = 9.82, *p* = 0.000 [<0.05]) and social anxiety scores (F = 8.74, *p* = 0.000 [<0.05]), but not in depression scores. Regarding the comparative mean scores, the current stage of online teaching in postgraduate anxiety equalization (44.35 ±11.36) and the postgraduate anxiety equalization of online teaching combined (43.38 ± 10.43) were the highest; participation line Postgraduate Anxiety Equalization under teaching was the lowest (42.06 ± 9.34); and postgraduate students who received online education showed the highest degree of social anxiety (13.34 ± 7.77). We believe that the new teaching method seems strange to most postgraduate students. Factors such as adapting to the rhythm of online learning and the problems with timely feedback from teachers during online learning may be associated with the high severity of anxiety and social anxiety [39].

#### 3.3.5. Communication Mode-Based Differences

The ANOVA analysis (see Table 4) showed that there were significant communication mode-based differences in the anxiety scores and the depression scores between the groups (F = 6.84, *p* = 0.000 [<0.05]; F = 6.9, *p* = 0.000 [<0.05]). After equalizing, the anxiety scores for face-to-face communication with advisors were the lowest (41.74 ± 9.38), and the depression equalization also had a variety of communicators, the lowest in the formula (46.65 ± 10.82). The analysis of variance for the scale scores of postgraduate students with different methods of communication with students also yielded the same results at this stage (F = 8.41, *p* = 0.000 [<0.05]; F = 6.73, *p* = 0.000 [<0.05]). Postgraduate students who communicated in a face-to-face manner with advisors showed the lowest equalized anxiety and depression scores. As a result, face-to-face communication with advisors or students could effectively relieve postgraduate anxiety and depression. During a situation such as that of the pandemic, postgraduate students who can only communicate with advisors or students using video calls, character communication, etc., face more serious problems of anxiety and depression than those who communicate face-to-face. There are significant communication mode-based differences in the social anxiety scores of postgraduate students. Through equalized comparison, it was found that the main communication mode is “character communication,”, i.e., postgraduate students who communicate with students and advisors through email, WeChat, and other methods have the highest social anxiety scores. It also underlines the need to address the increased interpersonal challenges faced by postgraduate students in this group. Furthermore, it highlights the need to adopt targeted measures such as psychological orientation, to encourage the diversity in forms of online communication, and to fully utilize multimedia technology and video and voice communication [40].

The three modes of communication, “telephone”, “video”, and “character communication” were collectively considered “indirect communication” and “face-to-face” was considered “direct communication”. A two-way ANOVA analysis was carried out using communication with the advisors and communication with peers as independent variables and the anxiety or depression scores as dependent variables. The results are shown in Table 5 and Table 6.

Table 5 shows that, for the anxiety score, the main effects of the modes of communication with advisors (F = 20.70, *p* < 0.05) and with peers are significant (F = 8.152, *p* < 0.05), and the interaction effect is also significant (F = 7.25, *p* < 0.05). Further, a simple effect test shows that, for the students communicating with advisors indirectly, there is no significant difference between them and the students communicating with advisors directly (F = 0.05, *p* = 0.830 [>0.05]). Those who could communicate directly with the advisors experienced a significantly higher level of anxiety than those who could communicate directly with peers (F = 14.74, *p* < 0.05). As shown in Table 6, for the depression score, the main effects of the modes of communication with the advisors (F = 20.341, *p* < 0.05) and with peers (F = 7.081, *p* < 0.05) and the interaction effect were all significant (F = 6.491, *p* < 0.05). Further, a simple effect test showed that those who could indirectly communicate with the advisors were less depressed. There is no significant difference between the students who communicated indirectly and directly (F = 0.15, *p* = 0.697 [>0.05]). The depression score was significantly higher for those who could communicate directly with their advisors than with their peers (F = 13.42, *p* < 0.05).

In summary, postgraduate students who could communicate with advisors and peers directly had the lowest anxiety and depression scores. This result shows that the direct communication with peers and advisors is essential to ensure the mental health of students. Teacher–student and student–student relationships are important factors that affect postgraduate mental health and are two key directions for providing psychological counseling to postgraduate students.

## 4. Discussion

### 4.1. Mental Health of Postgraduate Students

This study showed that during the pandemic, 20.91% and 33.88% of postgraduate students experienced anxiety and depression symptoms. Furthermore, postgraduate students experienced more serious social anxiety problems compared to the Chinese norms. Mental health problems in postgraduate students are more serious during the pandemic than before the pandemic. We believe that the pandemic and lockdown measures have exacerbated the negative feelings of anxiety, depression and social anxiety among postgraduate students, as can be seen in longitudinal studies in other countries [41,42,43].

In addition to cross-country comparisons, it is necessary to compare the mental health of the Chinese postgraduate population with the Chinese population. Ren et al. [44] conducted a meta-analysis of 12 studies that included 27,475 participants in mental health problems during the pandemic in China. The results showed that the aggregate rate of anxiety detection on the SAS scale was 14% and the depression detection rate was 28%. Compared with this study, the detection rate of anxiety and depression is significantly higher, which indicates that the anxiety and depression problems in postgraduate students are more serious than the general population, and that more attention needs to be paid to the mental health of postgraduate students.

We believe that this is related to the huge impact of the pandemic on postgraduate students’ lives [45,46]. Studies have shown that during the pandemic, public concerns become more prominent, and the country’s tight prevention and control measures also brought about a tense environment. Moreover, the pandemic had an impact on the economy and industry, and this hindered postgraduate internships and research work, which could also be sources of anxiety, depression, and other psychological problems [47].

### 4.2. Influence of Professional Background and Degree Type

Our comparative analysis revealed that anxiety and depression were more severe among academic and liberal arts postgraduate students than among those pursuing other majors and degree types, and this finding is similar to that reported by Wang [48] and Odriozola-González [24].

Academic postgraduate students experience more stress, maintain higher standards, and require more academic support than professional postgraduate students, and long-term isolation due to the pandemic increases the impact on their learning programs. Therefore, during the pandemic, we should consider the practical difficulties faced by academic postgraduate students, adjust the curriculum appropriately, and maintain communication and orientation with postgraduate students [47].

Liberal arts students experience more serious anxiety and depression, which may be due to the fact that liberal arts students are more sympathetic and emotionally sensitive, deal with more interpersonal issues, and are more susceptible to social and environmental circumstances.

Shi et al. [49] surveying postgraduate students with SCL-90 found that liberal arts students had more psychological problems than science students. Posselt et al. [50] found that liberal arts students were more likely to develop psychiatric related illnesses but seek treatment less often than other students. Based on this reality, colleges and universities should pay attention to the mental health of postgraduate students in the liberal arts, understand the specific plight of liberal arts students in a targeted manner, and provide more emotional care.

### 4.3. Influence of Communnication Modes

This study found that the mental health of postgraduate students was related to the way they communicate with advisors and students. This study believes that communication modes are closely related to social isolation [51]. Postgraduate students who are unable to communicate directly with advisors and students have relatively higher levels of anxiety, depression, and social anxiety. Postgraduate students who study at home cannot communicate directly with advisors and peers, cannot obtain direct and timely feedback in social interactions, and experience psychological distance. A study by Li and et al. [51] of 1805 Chinese respondents found that online social interactions, as represented by WeChat, actually had a negative effect on mental health. Consistent with Li’s findings, this study further illustrates the psychological irreplaceability of direct face-to-face communication.

Therefore, we should always pay attention to the mental health of postgraduate students as they study from home during the pandemic. Tutors should pay more attention to the life and studies of postgraduate students. They should also encourage postgraduate students to increase their communication with each other. Communication methods should not be limited to mail, WeChat, SMS, and other character communication methods, but these must include voice calls, video calls, and other ways to communicate more closely. Postgraduate students, under restricted conditions, experience difficulties in carrying out their out-of-school social activities. They should pay more attention to direct communication with advisors and peers, improve their academic understanding, and relieve stress using face-to-face communication.

### 4.4. Suggestions

In summary, in the face of the public health crisis of COVID-19, postgraduate students should be actively guided to help them realize that anxiety, depression, and other mental health problems are not due to illness or timeliness, but are normal responses by normal people in extraordinary times. Universities, advisors, counsellors, and other relevant personnel should employ the following guidelines: 

(1) Actively guide postgraduate students to correctly understand their mental health status and individual differences in mental tolerance and encourage postgraduate students to seek help if they experience psychological problems so as to help them adjust their goals and plans according to reality, and avoid the development of other problems such as post-traumatic stress disorder [31,52];

(2) Establish an early warning system for the mental health of postgraduate students during the pandemic and improve online and offline psychological counseling service systems;

(3) Consider the characteristics and situation of different postgraduate groups for postgraduate student management to develop targeted mental health education programs and adopt objective measures, so as to improve postgraduate mental health and nurture both physical and mental health to facilitate China’s modernization; 

(4) Develop and maintain conditions to improve the communication between postgraduate students and advisors during the pandemic and create a new postgraduate guidance mode to relieve the psychological problems of postgraduate students [53].

## 5. Limitations 

There are some limitations to our study that should be noted. The first limitation was the sampling technique used, which relied on digital infrastructure and voluntary participation, which increases selection bias. Second, the study was obtained from one specific area; our study design did not involve more psychological concepts, such as social distance, loneliness, risk perception, etc. Third, this study was a cross-sectional design of the survey, there was no follow-up period for the participants.

## 6. Conclusions

This research showed that during the pandemic, 20.91% and 33.88% of postgraduate students experienced anxiety and depression symptoms. Furthermore, postgraduate students experienced serious social anxiety problems. As a conclusion, the mental health of Chinese postgraduate students is significantly affected when faced with public health emergencies and lockdown measures. Besides, this study shows profession background, degree type and communication mode are the main influence factors of mental health. It is suggested that the government, schools and parents should collaborate to resolve this problem in order to provide high-quality, targeted psychological services to postgraduate students.

## Figures and Tables

**Table 1 ijerph-18-11542-t001:** Postgraduate Anxiety and Depression Description Statistics.

	Participants		Norms		*t*
	M	SD	M	SD	
SAS	42.60	9.85	29.78	5.46	72.91 ***
SDS	47.58	11.02	41.88	10.57	28.97 ***

*** *p* < 0.001.

**Table 2 ijerph-18-11542-t002:** Comparison of rates of anxiety and depression in other studies and this study.

Authors	Year	*n*	Research Objects	Rate of Anxiety	Rate of Depression
Wu X et al. [35]	2013	177	Graduate School of Medicine.	10.7%	24.3%
Wang Juan [36]	2017	941	Professional degree postgraduate students from 3 universities in Chongqing.	2.6%	8.5%
Chang J et al. [20]	2020	3881	Guangdong University Students.	26.60%	--
Mao S et al. [37]	2020	240	Medical postgraduate student of a university.	24.17%	38.75%
This study	2021	3137	China postgraduate students.	20.91%	33.88%

**Table 3 ijerph-18-11542-t003:** Social Anxiety Descriptive Statistics.

	Participant		Norm		*t*
	M	SD	M	SD	
Social avoidance	6.51	3.91	4.14	2.62	34.00 ***
Social distress	6.16	4.20	3.92	3.1	29.87 ***
Total SAD	12.67	7.71	8.03	4.64	33.73 ***

*** *p* < 0.001.

**Table 4 ijerph-18-11542-t004:** The score comparison of different types of postgraduate students (M ± SD).

Postgraduate Category		n	Statistics	Anxiety	Depression	Social Anxiety
Gender	Male	672		42.09 ± 10.15	46.92 ± 11.43	12.20 ± 7.76
	Female	2465		42.75 ± 9.77	47.76 ± 10.91	12.80 ± 7.69
			t	2.38	3.10	3.26
			*p*	0.123	0.078	0.071
Degree type	Academic	1142		43.08 ± 10.32	47.97 ± 11.24	13.11 ± 7.77
	Professional	1695		42.20 ± 9.42	47.25 ± 10.83	12.31 ± 7.64
			t	6.15	3.39	8.04
			*p*	0.013 *	0.066	0.004 **
Professional type	Science and engineering	456		42.92 ± 9.51	47.99 ± 11.26	12.90 ± 7.55
	Liberal arts	487		43.42 ± 10.25	48.98 ± 11.53	12.99 ± 7.45
	Social science	2194		42.36 ± 9.83	47.19 ± 10. 84	12.56 ± 7.80
			F	2.59	5.65	0.87
			*p*	0.075	0.004 **	0.419
Teaching method	Online teaching	253		44.35 ± 11.36	49.34 ± 11.66	13.34 ± 7.77
	Mixed teaching	864		43.38 ± 10.43	47.76 ± 11.46	11.76 ± 7.71
	Offline Teaching	2020		42.06 ± 9.34	47.28 ± 10.73	12.98 ± 7.67
			F	9.82	4.07	8.74
			*p*	0.000 ***	0.017 *	0.000 ***
Communication mode with advisors	Telephone	266		43.3 ± 10.08	48.50 ± 11.48	11.93 ± 7.54
	Face to face	1391		41.74 ± 9.38	46.65 ± 10.82	12.48 ± 7.69
	video	193		42.59 ± 10.22	47.07 ± 10.89	11.62 ± 7.81
	Character communication	1287		43.40 ± 10.18	48.87 ± 11.09	13.20 ± 7.72
			F	6.84	6.9	4.29
			*p*	0.000 ***	0.000 ***	0.005 **
Communication mode with peers	Telephone	119		43.19 ± 10.38	48.28 ± 11.67	11.46 ± 6.97
	Face to face	1154		41.73 ± 9.43	46.69 ± 10.67	12.55 ± 7.73
	Video	111		44.31 ± 10.74	48.76 ± 11.98	10.68 ± 6.91
	Character communication	1353		43.42 ± 10.12	48.44 ± 11.22	13.08 ± 7.7
			F	8.41	6.73	4.85
			*p*	0.000 ***	0.000 ***	0.002 **

* *p* < 0.05, ** *p* < 0.01, *** *p* < 0.001.

**Table 5 ijerph-18-11542-t005:** Two-way ANOVA results for anxiety.

	df	SS	MS	F	*p*
Communication mode with advisors	1	1924	1923.7	20.70	0 ***
Communication mode with peers	1	758	757.5	8.152	0 ***
Interaction	1	674	673.7	7.25	0 ***
Residual	2825	262,494	92.9		

*** *p* <0.001.

**Table 6 ijerph-18-11542-t006:** Two-way ANOVA results of depression.

	df	SS	MS	F	*p*
Communication mode with advisors	1	2401	2401.4	20.341	0 ***
Communication mode with peers	1	836	835.9	7.081	0 ***
Interaction	1	766	766.3	6.491	0.01 *
Residual	2825	333,507	118.1		

* *p* < 0.05, *** *p* < 0.001.

## Data Availability

The data are not publicly available due to privacy restrictions.

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
