# Peer review of "The Impact of the COVID-19 Pandemic on Chinese Postgraduate Students’ Mental Health"

_ijerph, 2021, doi:10.3390/ijerph182111542_

Round 1
Reviewer 1 Report
Dear authors,
The topic is of interest, the research is relevant and clearly presented.
There are however a few suggestions I would like to make.
- Include a literature review section to better illustrate and discuss the research topics and research results.
- The discussion section should be linked to the literature review section, the research results should be compared to the results obtained from other studies by other researchers or this research team.
- References should be included more often throughout the article, the information presented should be supported by evidence, i.e., references (articles, books, studies, project results, etc.). Throughout the article, explanations regarding the inclusion of reference in certain places are needed. Examples have been given in the attached reviewed article.
- Clarity should be considered here and there. Examples have been given in the attached reviewed article.
- Punctuation and spelling should be checked once more. Examples have been given in the attached reviewed article.
I would like to congratulate the authors for choosing to research this topic and the for the extensive survey, as the number of respondents is quite impressive.
Good luck!

Author Response
Dear reviewer,
Thank you for your valuable suggestions on this manuscript. We have modified it according to your suggestions. Relevant modifications and supplements are as follows:
Comment 1: Include a literature review section to better illustrate and discuss the research topics and research results.
Reply: We have summarized relevant literature in the Introduction section, in order to make our research results integrate with the related topic closer.
Comment 2: Include a literature review section to better illustrate and discuss the research topics and research results.
Reply: In the Introduction section and the Discussion section, we have referred to more research results from others to make clearer comparison and discussion.
Comment3: References should be included more often throughout the article, the information presented should be supported by evidence, i.e., references (articles, books, studies, project results, etc.). Throughout the article, explanations regarding the inclusion of reference in certain places are needed. Examples have been given in the attached reviewed article. Clarity should be considered here and there. Examples have been given in the attached reviewed article.
Reply: We have modified the problem about irregular references . and provided several proofs from existing researches and theories for our conclusions.
Comment 4.Punctuation and spelling should be checked once more. Examples have been given in the attached reviewed article.
Reply: We have invited a native English speaker to further check spelling and grammar, and refine the entire article.
Thank you again for your valuable suggestions. Your suggestions make our article better, and we are looking forward to your reply.
Yours sincerely,
Derong Kang.
Reviewer 2 Report
Thank you for the opportunity to review the manuscript entitled “The impact of the COVID-19 pandemic on Chinese postgraduate students’ mental health”.
Introduction:
The introduction section seems very long-winded. I strongly suggest that the authors shorten it and not dwell on it too much. Rather, they should give an overview in general terms of the areas of interest affected by lockdown. Why is the mental sphere important? sure, it affects a broad pattern of conditions, but this needs to be specified. A subject's lifestyle depends on his or her mental state, e.g., diet (10.3390/ijerph17197073) has been one of the dependent variables in this regard, and we know how much the covid (and stress associated with lockdown) has altered our lifestyle. These points need to be expanded upon in the introduction, which needs to be meaty yet concise.
Methods:
It is unclear how these students were selected. The authors need to more clearly specify the inclusion and exclusion criteria used.
No limitations of the study were reported. There are of course no studies without limitations, please specify.
The study protocol, although not an intervention, needs to be registered and the alphanumeric code entered in the methodology section.
Globally the work is interesting but needs to be improved and corrected
Author Response
Dear reviewer,
Thank you for your valuable suggestions on this manuscript. We have modified it according to your suggestions. Relevant modifications and supplements are as follows:
Comment1:
The introduction section seems very long-winded. I strongly suggest that the authors shorten it and not dwell on it too much. Rather, they should give an overview in general terms of the areas of interest affected by lockdown. Why is the mental sphere important? sure, it affects a broad pattern of conditions, but this needs to be specified. A subject's lifestyle depends on his or her mental state, e.g., diet (10.3390/ijerph17197073) has been one of the dependent variables in this regard, and we know how much the covid (and stress associated with lockdown) has altered our lifestyle. These points need to be expanded upon in the introduction, which needs to be meaty yet concise.
Reply: we have rewritten the Introduction section, deleted some of the content that is not highly relevant to the research topic, and re-discussed more researches in related fields. And we have further discussed the significance and related factors affecting postgraduate mental health during the COVID-19 pandemic, and simplified the Introduction section, and enriched the content structure.
Comment2:
It is unclear how these students were selected. The authors need to more clearly specify the inclusion and exclusion criteria used.
No limitations of the study were reported. There are of course no studies without limitations, please specify.
The study protocol, although not an intervention, needs to be registered and the alphanumeric code entered in the methodology section.
Reply:
(1) We have added the sampling process, sample selection and exclusion criteria.
(2) We have added the Limitations section (in Section 5) to discuss the limitations of this research.
(3) What needs to be explained is that in China, the study protocol and the informed consent agreement is a part of the ethical reviewing process, so the study protocol t has been covered in the alphanumeric code. And we have added it in the [Institutional Review Board Statement].
Thank you again for your valuable suggestions. Your suggestions make our article better, and we are looking forward to your reply.
Yours sincerely,
Derong Kang.
Round 2
Reviewer 1 Report
Dear authors,
The article looks much better now.
There are however three aspects I would like to highlight and that, in my opinion, still need to be addressed:
- Some sentences should be linked in one paragraph for coherence and cohesion reasons.
- Punctuation and the insertion of spaces should still be checked in the entire document.
- The journal’s referencing guidelines should be checked and observed in the article.
Examples have been given in the attached reviewed manuscript.
Otherwise, congratulations for the work you have done!
Kind regards,
The Reviewer

Author Response
Dear editor & reviewers,
Thank you for your valuable suggestions on this manuscript. We have modified it according to your suggestions. Relevant modifications and supplements are as follows:
For the reviewer 1:
Comment 1: Some sentences should be linked in one paragraph for coherence and cohesion reasons.
Reply: We have checked spelling and grammar, and made some change for text coherence and cohesion .
Comment 2: Punctuation and the insertion of spaces should still be checked in the entire document.
Reply: We have checked the punctuation and the insertion of spaces, and refine the entire article.
Comment3: The journal’s referencing guidelines should be checked and observed in the article.
Reply: According to the referencing guidelines ,we have modified the problem about references .
Reviewer 2 Report
Comment1:
The introduction section seems very long-winded. I strongly suggest that the authors shorten it and not dwell on it too much. Rather, they should give an overview in general terms of the areas of interest affected by lockdown. Why is the mental sphere important? sure, it affects a broad pattern of conditions, but this needs to be specified. A subject's lifestyle depends on his or her mental state, e.g., diet (10.3390/ijerph17197073) has been one of the dependent variables in this regard, and we know how much the covid (and stress associated with lockdown) has altered our lifestyle. These points need to be expanded upon in the introduction, which needs to be meaty yet concise.
References have not been revised in accordance with changes in the text. You should mention the work I referred to since you mentioned it in the new version.
Author Response
Dear reviewers & editor,
Thank you for your valuable suggestions on this manuscript. We have modified it according to your suggestions. Relevant modifications and supplements are as follows:
Comment1:
References have not been revised in accordance with changes in the text. You should mention the work I referred to since you mentioned it in the new version.
Reply:
I'd like to apologize for our carelessness.
Now we cite the article (10.3390/ijerph17197073) in our manuscript to make a discussion about how much the covid (and stress associated with lockdown) has altered our lifestyle. You can see it in the reference [7].
According to the referencing guidelines ,we have modified the problem about irregular references .
Yours sincerely,
Derong Kang.